# Using Representation Expressiveness and Learnability to Evaluate Self-Supervised Learning Methods

**Yuchen Lu**
*Mila, University of Montreal*

**Zhen Liu**
*Mila, University of Montreal*

**Aristide Baratin**
*SAIT AI Lab, Montreal*

**Romain Laroche**
*Microsoft Research*

**Aaron Courville**
*Mila, University of Montreal, CIFAR*

**Alessandro Sordoni**
*Microsoft Research, MILA*

**Reviewed on OpenReview:** *https://openreview.net/forum?id=BxdrpnRHNh*

## Abstract

We address the problem of evaluating the quality of self-supervised learning (SSL) models without access to supervised labels, while being agnostic to the architecture, learning algorithm or data manipulation used during training. We argue that representations can be evaluated through the lens of *expressiveness* and *learnability*. We propose to use the Intrinsic Dimension (ID) to assess expressiveness and introduce Cluster Learnability (CL) to assess learnability. CL is measured in terms of the performance of a KNN classifier trained to predict labels obtained by clustering the representations with $K$-means. We thus combine CL and ID into a single predictor – CLID. Through a large-scale empirical study with a diverse family of SSL algorithms, we find that CLID better correlates with in-distribution model performance than other competing recent evaluation schemes. We also benchmark CLID on out-of-domain generalization, where CLID serves as a predictor of the transfer performance of SSL models on several visual classification tasks, yielding improvements with respect to the competing baselines.

## 1 Introduction

Despite impressive recent progress in self-supervised learning (SSL) (Chen et al., 2020a; Caron et al., 2021; Grill et al., 2020; Caron et al., 2020; 2018; He et al., 2020; Chen & He, 2021), the problem of properly evaluating the quality of the learned representations *without using labelled data* has not been fully explored. This is an important problem: solving it could give us a practical tool to choose which SSL model to use when downstream task labels are not accessible, or when the costs of fine-tuning every model to choose the best one are prohibitive. It can also shed light on how these methods work.

In this paper, we approach it by drawing an analogy between unsupervised learning and the evolution of human language. It has been suggested in the language evolution literature (see e.g., Smith et al., 2013) that linguistic structure results from competing pressures for expressiveness (to discriminate objects and concepts

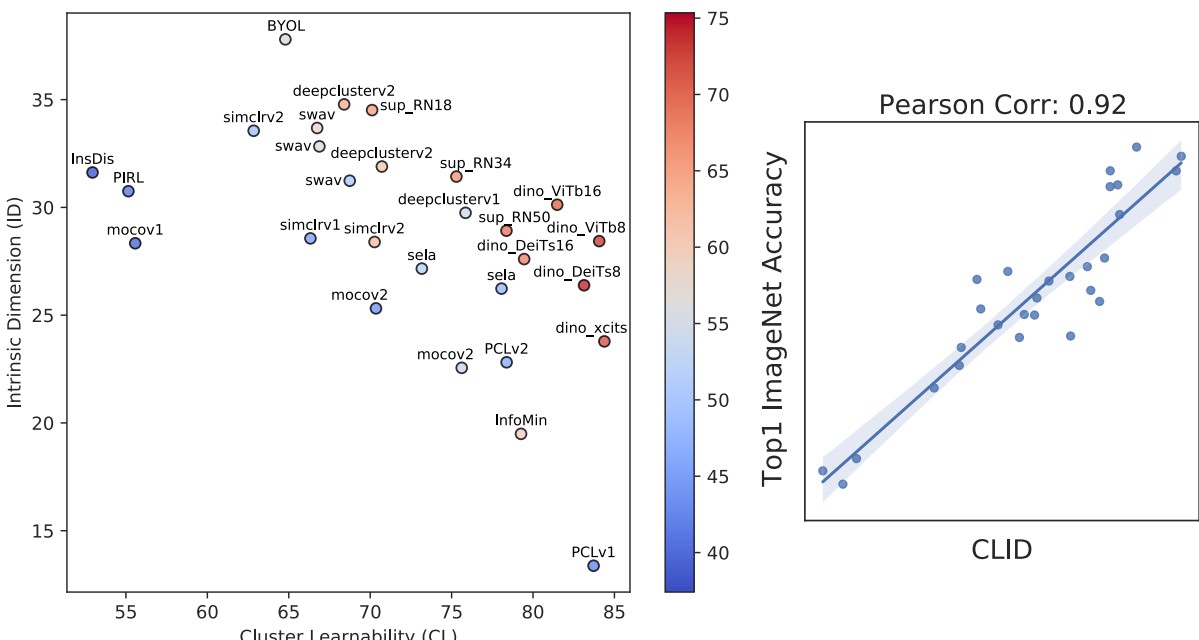

Figure 1: We propose to use Cluster Learnability (CL) to measure learnability and Intrinsic Dimension (ID) for expressiveness. **Left**: Each circle is a SSL pre-trained checkpoint, and the color is used to show its KNN Top1 ImageNet accuracy. Red indicates high accuracy, while blue indicates low accuracy. Good self-supervised learning representations are learnable and expressive, distributed over the upper-right portion of the graph. **Right**: Our predictor is highly correlated with ImageNet performance, even without access to gold labels and by staying agnostic to the model's architecture or training algorithm.

of the world) and learnability (to be transmitted across generations). Here, we take the view that a similar guiding principle may be applied to representations of natural images. Just as a language associates objects and scenes with utterances, SSL algorithms do it with continuous vectors to images. We hypothesize that *a good representation should not only cover the diverse visual concepts in the datasets, but also compress them in a manner that could be efficiently learned*. Our goal is to test this hypothesis by (*i*) proposing tractable metrics to assess *expressiveness* and *learnability*, (*ii*) designing performance predictors based on these, and (*iii*) testing these predictors through a large-scale empirical study of a diverse class of SSL methods.

Concretely, we propose to quantify expressiveness via an estimator of the intrinsic dimension (ID) of the data representations (Facco et al., 2017; Ansuini et al., 2019). To assess learnability, we take inspiration from Laina et al. (2020) and propose a novel method called Cluster Learnability (CL), based on the performance of a KNN learner trained to predict labels induced by clustering held-out representations with $K$-Means. We show that a combination of CL and ID, dubbed CLID, correlates with the model performance across different architectures (see Figure 1), better than existing evaluation scheme baselines (Wang & Isola, 2020; Reed et al., 2021; Yu et al., 2020). To further demonstrate the usefulness of our framework, we conduct out-of-domain generalization prediction experiments on seven downstream transfer tasks. We find that the proposed CLID predictor outperforms baseline concurrent methods at predicting transfer performance.

Our contributions are summarized below:

- We propose an evaluation framework for self-supervised learning relying on expressiveness and learnability, yielding new insights into existing techniques in the literature.

- We propose tractable and generic metrics to quantify *(i)* expressiveness via the Intrinsic Dimension (ID) and *(ii)* learnability via our novel Cluster Learnability (CL) metric.

- We show that CLID predictors are well correlated with Top-1 KNN classification accuracies on ImageNet, and are robust with respect to the choice of hyperparameters.

- We show that CLID predictors predict the transfer performance of pretrained SSL models on several visual classification task, especially when used jointly with in-domain accuracies.

## 2 Learnability and Expressiveness

In this section, we briefly discuss existing evaluation schemes of self-supervised learning methods, which will be used as baselines in our experiments. We then describe our own evaluation scheme, based on specific metrics to quantify learnability and expressiveness.

**Notation**  We consider the following setting: we assume we have an unlabelled dataset $\{x_i\}_{i=1}^N$ represented in $\mathcal{X} \subset \mathbb{R}^d$, where $x_i \sim \mathcal{P}$ are sampled i.i.d. from some distribution $\mathcal{P}$ over $\mathcal{X}$. We also consider *representation maps* $\mathcal{F} : \mathcal{X} \to \mathbb{R}^m$, typically pretrained neural networks, which represent any input as an $m$-dimensional vector. Any such map forms a representation dataset $Z = \{z_i\}_{i=1}^N$ where $z_i = \mathcal{F}(x_i)$.

### 2.1 Existing SSL Evaluation

**Alignment and Uniformity (Wang & Isola, 2020)**  By decomposing contrastive learning loss, Wang & Isola (2020) proposes to understand SSL with alignment and uniformity. Formally, they introduce two metrics,

$$\mathcal{L}_{align} := \mathbb{E}_{x \sim \mathcal{P}, x' \sim \mathcal{P}_{\mathrm{aug}}(\cdot|x)} \|\mathcal{F}(x) - \mathcal{F}(x')\|_2^\alpha, \quad \alpha > 0$$

$$\mathcal{L}_{unif} := \log \mathbb{E}_{x_1, x_2 \sim \mathcal{P}} \left[ e^{-t||\hat{\mathcal{F}}(x_1) - \hat{\mathcal{F}}(x_2)||_2^2} \right], \quad t > 0 \tag{1}$$

where $\mathcal{P}_{\mathrm{aug}}$ is the conditional distribution defined by the data augmentation procedure, $\hat{\mathcal{F}}(x) := \mathcal{F}(x)/\|\mathcal{F}(x)\|$ are the normalized features and $t, \alpha$ are tunable parameters. These two metrics respectively correspond to the intuition of learnability and expressiveness. By making the feature of positive pairs to be similar, the minimization of $\mathcal{L}_{align}$ induces learnable representations that can generalize from one positive example to another. Minimizing the other component $\mathcal{L}_{unif}$ ensures that the (normalized) features are distributed over a hyper-sphere, so that the representations can cover more concepts and achieve high expressiveness.

**Maximal Coding Rate Reduction (MCR2) (Yu et al., 2020)**  MCR2 measures the reduction of the average coding length per sample, a.k.a. the coding rate, of a representation, induced by the knowledge of some category structure (e.g., the membership of the samples in different classes). The goal of this approach is to learn representations that discriminates between classes while being maximally diverse (MCR2 corresponds to the intuition of learning a diverse and yet compressible (w.r.t. the class partition) representations.

**Mutual Information (MI) and Pretext Tasks**  Maximizing mutual information between inputs and representations (Oord et al., 2018; Bachman et al., 2019a) can be viewed as increasing the expressiveness of the representations. However, as pointed out by Tschannen et al. (2020), MI is not a good predictor of the model performance. Our view is that this is because MI alone does not cover take into account the learnability of the representations. Similar arguments can be applied to pretext tasks evaluation scheme like rotation prediction and solving jigsaw puzzles (Reed et al., 2021).

### 2.2 Our Proposal

Here we describe our evaluation scheme under the view of learnability and expressiveness. Our proposed metrics can be efficiently evaluated on the validation set of the dataset[1].

---

[1]We have also experimented with metrics computed on the train set but we observed no significant difference in the results.

**Intrinsic Dimension (ID)**   We propose to use the notion of intrinsic dimensionality (ID) of the data in the representation space (Pettis et al., 1979a) to quantify expressiveness. Our intuition is that, as more and more fine-grained categories emerge in the representation space, we expect the manifold complexity to increase. Intrinsic dimension is a way to quantify this complexity, characterized as the number of parameters needed to describe the representation manifold without loss of information.

Inferring the intrinsic dimension of a highly nonlinear manifold is a challenging problem (e.g., Levina & Bickel, 2005). In this work, we leverage the nearest neighbor-based method of Facco et al. (2017) to estimate ID[2]. This estimator (TwoNN) is shown to be reliable with respect to representation dimensions and scalable to real-world datasets with deep neural networks (Ansuini et al., 2019). The concept of intrinsic dimension is also connected to entropy (we discuss this point in detail in Appendix B).

Formally, given $z_i \in Z$ and an integer $k \geq 1$, we denote by $r_{ik} = D(z_i, NN(z_i, k))$ the distance[3] of $z_i$ to its $k$-th nearest neighbor $NN(z_i, k)$. Assuming that the points are sampled on a manifold with intrinsic dimension $d$, it can be shown that, under the assumption of local uniformity[4], the ratio of distances $\mu_i := r_{i2}/r_{i1}, 1 \leq i \leq N$, follow a Pareto distribution with parameter $d+1$ on $[1, \infty)$, i.e., $\mu_i \sim P(\mu|d) := d\mu^{-(d+1)}$. While $d$ can be computed by maximizing the likelihood, we follow a much simpler method proposed by Facco et al. (2017) based on the cumulative distribution $F(\mu) = 1 - \mu^{-d}$ associated with $P(\mu|d)$. The idea is to estimate $d$ with a linear regression on the empirical cumulate of the distribution. Specifically, assuming we sort $\mu_i$ ascendingly, that is $\mu_1 \leq \mu_2 \leq ... \leq \mu_N$, we estimate the cumulative distribution as $F_i^{emp} = i/N$ and fit a straight line on the datasets $\{(\log \mu_i, -\log(1 - F_i^{emp}))\}_{i=1}^N$ in the two dimensional plane. The slope is the estimated ID.

**Cluster Learnability (CL)**   Let $\{z_i, \tilde{y}_i\}_{i=1}^N$ be the labelled dataset obtained by clustering the representation (e.g., with $K$-means). We define the *learnability* of the representation as the performance of a classifier on this labelled dataset. For practical purpose, we choose KNN classifier due to its efficiency. We re-use the dataset split to assess the performance of a KNN classifier on this labelled dataset. Let $\hat{y} = KNN(z; \{z^{train}, \tilde{y}^{train}\})$ be the prediction of $z$ after seeing the training data. Learnability is defined as:

$$CL = \frac{1}{N} \sum_{i=1}^N [\hat{y}_i^{val} == \tilde{y}_i^{val}] \tag{2}$$

**CLID Predictor**   We apply our CL and ID in the context of predicting models performance ranking. As a result, for each checkpoint model, we combine the computed CL and ID values into a single numeric predictor, which can be used to rank the models. To deal with the different numeric scales of CL and ID, we firstly standardize them across models into a range from 0 to 1 and then adding these values together:

$$\text{CLID}: \quad CL + ID.$$

Furthermore, if we have access to the in-domain performance of the models, we can learn a weighted sum

$$\text{W-CLID}: \quad \mathbf{w}^\top [CL, ID]$$

where $\mathbf{w} \in \mathbb{R}^2$ is a weight that can be learned by linear regression on the in-domain performances. This predictor is mainly used when predicting out-of-domain performances.

---

[2]We experiment with the ID estimator based on the maximum likelihood estimation (Levina & Bickel, 2004; Ma et al., 2018), without noticing a difference in performance.

[3]While we generally consider nearest neighboors w.r.t Euclidean distance, in practice we will also use the cosine distance function, $D(z_1, z_2) = 2 - 2\cos(z_1, z_2)$. Both produce similar results in our experiments.

[4]While the original derivation (Facco et al., 2017) assumes global uniformity, a post-hoc analysis shows that it only needs local uniformity up to the second neighbor.

## 3 Empirical Study

### 3.1 Setup

We select in total 28 self-supervised learning checkpoints trained on ImageNet over different algorithms, architecture, and training epochs. A complete list can be found in Table 4 in the appendix. We use the KNN evaluation on the validation data using the ground-truth labels to measure the performance of the model, which has been shown to be well correlated with the linear evaluation but computationally less expensive (Caron et al., 2021).

For the computation of cluster learnability, we choose the square root of the dataset size as the number of clusters in Kmeans. We report results with 1 neighbor for our KNN learner. We also show that our results are robust to other choices of hyper-parameters. We normalize the features and use cosine distance for the K-means clustering and KNN learner[5]. Other configurations of cluster numbers and neighbor numbers are also explored in section 3.4. All our experiments are computed on a single V100 GPU.

### 3.2 Baselines

Wang & Isola (2020) also proposes to predict the ImageNet performance of pre-trained SSL checkpoints as an evaluation scheme. Therefore, in our experiments, we follow the official implementation[6] with $\alpha = 2$ and $t = 2$ as default values for the tunable parameters in Eqn 1. We additionally define $-\mathcal{L}_{contrast} = -\mathcal{L}_{align} - \mathcal{L}_{unif}$ to compute the predictor for Wang & Isola (2020)'s method, which reduces to the negative contrastive loss. We add a negative sign, since we require a predictor to be in proportional to the model performance.

The original MCR2 (Yu et al., 2020) requires a class partition to be pre-specified. In order to adapt it into an unsupervised evaluation scheme, we use a $K$-means clustering as the dataset partition. We follow the default settings[7] and we normalize the features. We also experiment with using the ground-truth label as the dataset partition, but we found no improvements. We use the coding rate reduction $\Delta R$ (see Yu et al., 2020, Equation 6) to be maximized as a predictor.

For the Mutual Information baseline, we use MINE (Belghazi et al., 2018) with a fixed student ResNet18 network (He et al., 2016) to estimate the mutual information between inputs and representation. We use a batch size 128, learning rate 0.0005 and weight decay 0.001. The network is trained for 50000 steps on the training images, and we report MINE on the validation data. We find that training longer is computational intensive, while the results are similar.

For the pretext task, we experiment with rotation prediction by constructing a 4-way classification. We randomly rotate the training images by $0, 90, 180, 270$ degrees, train a KNN classifier on the training images to predict a 4-way classification, and then report the rotation prediction accuracy on the validation images. This is shown to be an effective evaluation in both self-supervised learning (Reed et al., 2021) and architecture search (Liu et al., 2020).

### 3.3 Is CLID scheme correlated with ImageNet performance?

We first investigate whether the proposed evaluation scheme is useful for in-domain (ImageNet) generalization. We perform both qualitative and a quantitative examination and show the results in Figure 1. On the left, we find that the self-supervised learning checkpoints with higher ImageNet accuracies tend to be both more learnable and more expressive (e.g., in the upper-right corner of the graph). In the meantime, we find that methods favouring only of these qualities at the expanse of the other, like PCLv1 and PIRL, also have poor ImageNet accuracies. In Figure 1 middle and right, we compute the correlation of our CLID predictors with respect to the ImageNet accuracy of the model considered and show that we achieve a Pearson $\rho$ of 0.92 for CLID.

---

[5]We find similar results when using L2 distance.
[6]See https://github.com/SsnL/moco_align_uniform
[7]See https://github.com/ryanchankh/mcr2

Table 1: Correlation results between ImageNet performances and different predictors. We compute both Pearson $\rho$ and Kendall $\tau$. Our CLID predictors achieve the highest correlation.

| Predictors | Pearson $\rho$ | Kendall $\tau$ |
|---|---|---|
| $-\mathcal{L}_{align}$ | 0.42 | 0.26 |
| $-\mathcal{L}_{unif}$ | -0.05 | 0.03 |
| $\mathcal{L}_{contrast}$ | 0.37 | 0.24 |
| $\Delta R$ | -0.62 | -0.33 |
| MI | 0.13 | 0.08 |
| Pretext | 0.61 | 0.27 |
| *Ours* | | |
| CL | 0.74 | 0.44 |
| ID | 0.12 | 0.09 |
| CLID | **0.92** | **0.75** |

In Table 1, we compare our results with the baselines. The regression plots for the baselines can be found in Figure 3. Our proposed predictors achieve the highest correlation both in terms of Pearson $\rho$ and Kendall $\tau$ coefficient. We find that $\mathcal{L}_{contrast}$ achieves a relatively low correlation $\rho = 0.37$, which apparently contradicts the observation made in Wang & Isola (2020). We hypothesize that this is due to the fact that their analysis is restricted to SSL checkpoints trained with contrastive learning, and therefore $\mathcal{L}_{contrast}$ might not be general enough to characterize representations obtained by alternative approaches. Additionally, the results in Wang & Isola (2020) are computed on the representations used to compute the noise-contrastive objective, which are usually transformed by a final projection layer that is not always present in other SSL methods. This limits its generality. Interestingly, we find that MCR2 ($\Delta R$) gives negative correlation which warrants further investigation. While the MI baseline only achieves $\rho = 0.13$, the pretext task baseline is still surprisingly good with a $\rho = 0.61$, suggesting that the rotation prediction task remains a simple but effective baseline.

### 3.4 Is CLID sensitive to its hyper-parameters?

In this section, we check the sensitivity of CLID to its hyperparameters. Since ID does not have any hyper-parameters, we mainly examine the following hyper-parameters for CL: the number of clusters and the number of neighbors used in the KNN Learner.

The results can be found in Figure 2. The resulting predictors can still produce a higher correlation than the rotation prediction for a wide range of the configurations we tried. We observe that decreasing the number of neighbors leads to a better predictor. Given our result, we recommend setting this number to be 1.

We observe that there is an optimal cluster number so that the choice of neighbor numbers can be more flexible. If the cluster number is too low, the results become worse. Recall that for different checkpoints, the clusters are produced with the same $K$-means algorithm, so the separability among clusters might be roughly controlled. As a result, we hypothesize that the number of clusters might control the underlying difficulty of the classification problem faced by the KNN learner. In the extreme case, we either have 1 cluster or as many clusters as data points. In the former, KNN accuracy would always be 100% and in the latter always be close to 0%[8]. As a result, neither of them would be suitable enough to distinguish the collected checkpoints. As a rule of thumb, we recommend using a reasonably large number of clusters, e.g., the square root of the dataset size.

### 3.5 Can CLID be used to predict transfer performance?

Here we investigate whether CLID can be a good predictor of the performance on downstream tasks. We collect 7 out-of-domain downstream visual classification tasks. For each domain, we compare the ranking of our SSL checkpoints induced by CLID, and the ranking induced by the actual test performance on that

---

[8]To be specific it's $1/N$, where $N$ is the dataset size. It's close to 0% when $N$ is large.

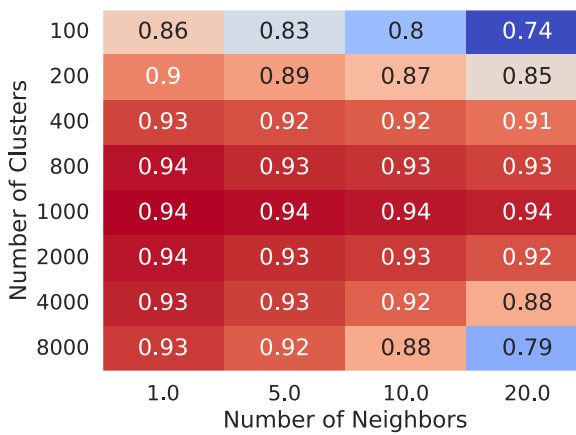 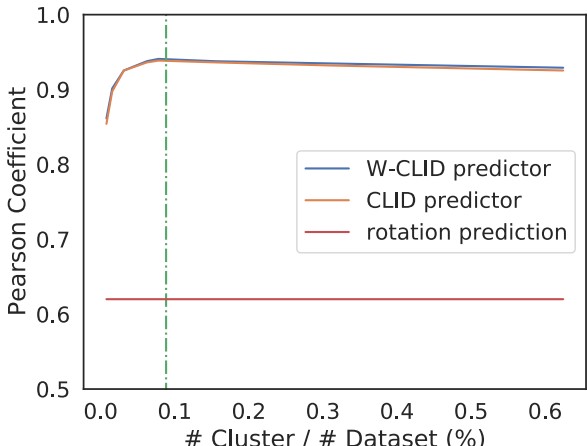

Figure 2: Robustness Analysis for ImageNet. **Left**: The heat map of Pearson Coefficient between the Top-1 accuracy and W-CLID predictor. **Right**: Pearson coefficient vs. the ratio between the number of clusters and the dataset size when using one neighbor. The result is stable with a reasonably large number of clusters, e.g., the square-root of the dataset size **(Green Dashed Line)**.

Table 2: Kendall Ranking Coefficient results on Out-of-Domain Tasks when ImageNet labels are not available. The CLID predictor has the best predictive performance for transfer tasks. Full results in Table 5, including results for the few-shot experiments.

|  | CLID | $-\mathcal{L}_{contrast}$ | $\Delta R$ | MI | Pretext |
|---|---|---|---|---|---|
| **foods** | **0.75** | 0.17 | -0.35 | 0.01 | 0.51 |
| **flowers** | **0.81** | 0.25 | -0.35 | 0.14 | 0.46 |
| **pets** | **0.71** | 0.23 | -0.25 | -0.08 | 0.54 |
| **caltech101** | 0.56 | 0.11 | -0.18 | -0.05 | **0.67** |
| **stl10** | **0.56** | 0.02 | -0.34 | -0.21 | 0.41 |
| **aircraft** | **0.67** | 0.21 | -0.27 | 0.16 | 0.57 |
| **cars** | **0.81** | 0.32 | -0.28 | 0.16 | 0.48 |
| **Avg** | **0.70** | 0.19 | -0.29 | 0.02 | 0.52 |

domain. We report the Kendall $\tau$ correlation score, which is consistent with the existing benchmarks on out-of-domain generalization (Vedantam et al., 2021).

We first investigate the scenario where the ImageNet labels are not accessible (Table 2). This is an extension of our previous scenario, and it is also a realistic assumption especially when the models are pretrained on web-scale data without clear annotations. We find that our CLID predictor is the best, even beating the W-CLID predictor. One reason is because the linear coefficients **w** are obtained from regressing in the ImageNet domain, which might increase overfitting and hurt transfer. We also find that $-\mathcal{L}_{contrast}$ is not a good predictor, and a simple pretext task like rotation prediction already has reasonable performance even in transfer settings.

In the second setting, we assume to have access to ImageNet labels (Table 3) and explore whether transfer performance prediction can be improved using this additional information. We find that the ImageNet accuracies have pretty high correlation with the downstream tasks, with an average Kendall coefficients of 0.71. To integrate ImageNet performance information to each our competing predictors, we proceed as follows. Let $r^i_{pred}$ be the ranking of $i$-th checkpoint from the predictor, and $r^i_{img}$ be that from the ImageNet accuracy.

Table 3: Kendall Ranking Coefficient results on Out-of-Domain Tasks when ImageNet labels are obtained. The results are computed with the *joint rank product* between the predictors and the ImageNet accuracy. While the ImageNet accuracy can predict the transfer accuracy reasonably, the proposed CLID predictor further enhances the predictive performance. Full results in Table 6, which include results with few-shot experiments. Blue cells indicates the joint ranking outperforms using ImageNet accuracy alone.

| | Imagenet | CLID | W-CLID | Source Ent | Target Ent | $L_{contrast}$ | Pretext |
|---|---|---|---|---|---|---|---|
| **foods** | 0.86 | 0.84 | 0.81 | **0.88** | 0.84 | 0.56 | 0.75 |
| **flowers** | 0.70 | **0.79** | 0.74 | 0.74 | 0.71 | 0.58 | 0.64 |
| **pets** | 0.75 | 0.77 | **0.78** | 0.66 | 0.72 | 0.55 | 0.76 |
| **caltech101** | 0.62 | 0.59 | 0.58 | 0.60 | 0.66 | 0.41 | **0.77** |
| **stl10** | **0.76** | 0.71 | 0.71 | 0.61 | 0.71 | 0.36 | 0.64 |
| **aircraft** | 0.60 | 0.66 | 0.61 | **0.68** | 0.64 | 0.50 | 0.62 |
| **cars** | 0.68 | **0.79** | 0.74 | 0.72 | 0.74 | 0.64 | 0.66 |
| **Avg** | 0.71 | **0.74** | 0.71 | 0.70 | 0.72 | 0.52 | 0.69 |

We compute a joint ranking by computing the rank product, which is the geometric mean of these two ranks:

$$r_{joint}^i = \sqrt{r_{pred}^i r_{img}^i}$$

In addition to comparing with previously presented baselines, we also add the strongest baselines from (Vedantam et al., 2021): we train a linear classifier and measure the negative label entropy $-H(Y|X)$. Since the entropy can be measured on either source domain or target domain, we denote each variant as "Source Ent" and "Target Ent". The underlying intuition is that, if the model has more confidence in the prediction (lower entropy), it should generalize better.

We find that the proposed CLID predictor can further enhance the ImageNet accuracies predictions, with an average Kendall coefficients of 0.74 for the joint ranking. We find that "Target Ent." and "Source Ent." both have a reasonable performance, which is consistent with observation in Vedantam et al. (2021), but they underperform our predictor. Note that computing "Target Ent." and "Source Ent." requires training an extra linear layer, which increases their computational requirements.

## 4   Related Work

**Representation Evaluation**   Several recent works address the question of representation evaluation in self-supervised learning. Whitney et al. (2021) propose to use the learning dynamics of the downstream classifiers to measure the representation complexity. However their method still depends on extra human labels. Pretext tasks like jigsaw or rotation prediction are shown to be well correlated with the self-supervised evaluation (Reed et al., 2021; Deng & Zheng, 2021) and architecture search (Liu et al., 2020). However, they also rely on crafting ad-hoc data augmentations. Ericsson et al. (2021) argues that it is important to look at the transfer performance of the SSL models in order to judge its quality. We agree with this statement and therefore also test our method for out-of-domain generalization settings (section 3.5). Closely related to our work, a very recent paper Garrido et al. (2022) builds on theoretical insights of He & Ozay (2022) to propose an evaluation of joint-embedding self-supervised methods without access to supervised labels based on the effective rank of the learned embedding matrix. While Garrido et al. (2022) employs the spectral rank of the representation as a metric, akin to measuring effective dimension, our paper approaches the problem from the perspective of performance across a tradeoff between effective dimension and learnability.

The concepts of learnability and expressiveness, while not being explicitly mentioned, can be found in existing literature. For example, many SSL strategies emphasize maximizing the mutual information between inputs and representations (Arora et al., 2019; Vincent et al., 2008; Higgins et al., 2017; Oord et al., 2018; Bachman et al., 2019b). These can be viewed as attempts to increase expressiveness, since high mutual information leads to correspondence between inputs and representations, and thus the visual concepts among the inputs

are mapped to the representation space. Recent works also pay attention to the emerging properties of learnability in the cluster structure from SSL models through human studies (Laina et al., 2020). The emphasis of learnability can be also found in the recent attempt to design a compression regularizor called Conditional Entropy Bottleneck (Lee et al., 2021). Finally, the intuition of a trade-off between learnability and expressiveness underpins other representation analysis frameworks, such as alignment and uniformity (Wang & Isola, 2020) or Maximal Coding Rate Reduction (Yu et al., 2020), which are further explained in Section 2. In this paper, we aim at turning this intuition into a practical self-supervised evaluation tool, especially on predicting performances in out-of-distribution transfer tasks.

**Learnability, Ease-of-Transmission and Compression**   Learnability has been argued to be a hallmark of the human language in order to be effortlessly transmitted through generations  (Kirby et al., 2014; Rafferty et al., 2011; Beckner et al., 2017; Zhou & Yurovsky, 2021; Kampen, 2004), and it is also true for visual concepts like color (Xu et al., 2010), categories (Griffiths et al., 2006), shapes (Portelance et al., 2021) etc. In deep learning, it has been explored in the context of emergent communication (Ren et al., 2020; Guo et al., 2019; Li & Bowling, 2019), language drift (Lu et al., 2020), and neural module networks (Vani et al., 2021), but it is less explored for vision representation learning, except for a human study on just two SSL methods (Laina et al., 2020). While Lee et al. (2021) also investigates a regularizor for compressive self-supervised learning, it is unclear whether such notion is useful as a representation evaluation framework. Learnability has a tight connection to compression (Chaitin, 2007) and prequential codelength (Dawid, 1984), which quantifies the compression levels with the online learning error. Existing work (Blier & Ollivier, 2018) has uses it to support the generalization ability of the learner (e.g., deep neural nets) on the dataset (e.g., labeled images). Here, we use this concept to quantify the learnability of the representation, in the sense that if the emerged Kmeans clustering is more learnable, then the same KNN learner could achieve a lower compression bound via prequential coding.

**Manifold Intrinsic Dimension**   Intrinsic dimension can be thought of as the smallest number of variable needed to approximate the representation manifold. Applying local neighborhood information to estimate the intrinsic dimension is not a new idea, and it was shown to be more efficient than the global eigenvalue approach (Pettis et al., 1979b). Ansuini et al. (2019) apply the TwoNN estimator (Facco et al., 2017) to the non-linear representation manifold of modern deep neural nets. They find that the intrinsic dimension is inversely correlated to the classification accuracy Their work is further extended to confirm that natural images lies in a low-dimension manifold (Pope et al., 2021), and that lower ID datasets leads to better generalization. Recanatesi et al. (2019) and and Li et al. (2018) further highlight the connections between intrinsic dimension and the generalization properties of learned representations, by comparing the models before and after supervised training. Intrinsic dimension can also be estimated locally with a Maximum Likelihood Estimator (Levina & Bickel, 2004), which Ma et al. (2018) propose to use as a regularizor against overfitting noisy labels. While the above work mainly focus on ID with supervised learning models, our work on SSL models presents a more nuanced view about ID. It is indeed the case that lower intrinsic dimension can lead to better accuracy, especially among supervised checkpoints (see "sup_RN18", "sup_RN34", "sup_RN50"in Figure 1). However, we hypothesize that when learning representation from scratch without labels, e.g., self-supervised learning, the representation manifold still need a certain amount of complexity in order to include enough information from the dataset.

## 5   Conclusion

We propose a unifying view to evaluate self-supervised representation learning through expressiveness and learnability. We propose to estimate expressiveness with intrinsic dimension (ID), and learnability with performance of a KNN learner on the K-means clustering of the representation. We show that the proposed CLID evaluation scheme better predicts the ImageNet accuracy than other evaluation schemes. We further demonstrate that our CLID can also predict the transfer task performance.

While our evaluation scheme is designed to be general, a potential limitation is that it might be only suitable to the classification tasks, since we put emphasis on the emergence of learnable cluster structure. As a result, the proposed predictor might not be as useful for other downstream tasks like object detection or

segmentation. While our results is true for a population of models, there are also some outliers. E.g., in Figure 1 "deepclusterv1" seems to have higher CL and ID than "simclrv2", but its accuracy is lower.

Future works could further explore the expressiveness-learnability in a more theoretical context, extend this framework to other field like pretrained language models, as well as devise new SSL algorithms that directly maximize intrinsic dimension or cluster learnability.

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
