# OpenReview forum: "Using Representation Expressiveness and Learnability to Evaluate Self-Supervised Learning Methods"
_TMLR — Accepted by TMLR_

### Review · Reviewer_SmAV · 2023-07-24

**Summary Of Contributions:**

This work investigates the problem of evaluating the quality of self-supervised learning (SSL) models. The typical way to evaluate such models is to measure the accuracy of linear classifiers of embedding vectors, but this requires a suitably large evaluation dataset with supervised (image, label) pairs, such as ImageNet. When there is no natural evaluation dataset or when labels are unavailable this approach is not possible, and the current work aims to develop a metric that can assess SSL model quality using only its unsupervised training data. Ideally, this metric would accurately predict the performance of linear probe classifiers without requiring supervised labels. The proposed metric has two components: Intrinsic Dimension, which is meant to measure the expressiveness of the model, and Cluster Learnability, which is meant to measure how easily concepts can be learned from the representations. ID is calculated using the TwoNN method from prior work, and CL is calculated by learning a K-means model of the representations, learning a KNN classifier from the K-means model results, and measuring the agreement between the K-means predictions and the KNN predictions on validation data. The final proposed metric is the sum of these scores after normalization. Experiments are conducted on a variety of pretrained SSL models which show the proposed metric can accurately predict ImageNet performance with linear probing, and that it can accurately predict OOD transfer performance with and without use of ImageNet labels.

**Audience:**

Yes

**Broader Impact Concerns:**

The authors did not discuss Broader Impacts, but this did not impact my evaluation of the work.

**Claims And Evidence:**

Yes

**Requested Changes:**

Could the authors provide a direct comparison between their work and existing results like Wang&Isola 2020 and MCR2 by applying their method to a scenario explored in the original papers?

**Strengths And Weaknesses:**

**Strengths**

* The research problem investigated in this work is a very relevant and interesting direction. Better tools for assessing SSL checkpoint quality without a canonical downstream supervised classification task could enable more reliable SSL evaluation in diverse domains, and provide speedier evaluations without model fine-tuning even when a downstream supervised classification task exists.
* The proposed method is lightweight and scalable. Both ID and CL calculations only require Euclidean distance measures (or other distance measure) and rankings in the activation space.
* Experimental results show strong correlation between the proposed metric and the typical metric of linear probe evaluation on Imagenet, and the proposed metric significantly outperforms other metrics studied.
* The method is intuitively reasonable, and the paper is clearly written and easy to follow.

**Weaknesses**

* The baseline methods such as (Wang&Isola 2020) and MCR2 are reimplemented for ImageNet, but no direct comparison is made to experiments present in the original papers. Since neither paper directly investigates ImageNet, it might be good to also apply the proposed method on a smaller-scale setting such as CIFAR-10 so that direct comparison to published results is possible.

---

> ### Author Response · Authors · 2023-08-31
> **Thank you for your positive feedback!**
>
> > The baseline methods such as (Wang&Isola 2020) and MCR2 are reimplemented for ImageNet, but no direct comparison is made to experiments present in the original papers.
>
> Your point is well-taken. Now, our decision to upscale to ImageNet is, in our view, a significant advancement, which aligns better with practical real-world scenarios. Our experimental setup does indeed facilitate direct comparison of the methods, even if the specific outcomes might not align precisely with the published results from the original papers.

---

### Review · Reviewer_gRum · 2023-07-25

**Summary Of Contributions:**

This paper aims to design an evaluation metric for self-supervised representations. To this end, the paper suggests measuring the expressiveness and learnability, which are estimated by intrinsic dimension (ID) and cluster learnability (CL), respectively. The model with better Pareto optimality tends to exhibit higher classification accuracy. Additionally, by combining them into a single metric, the proposed CLID metric shows a strong correlation with ImageNet accuracy.

**Audience:**

Yes

**Claims And Evidence:**

Yes

**Requested Changes:**

See [W1-3].

**Strengths And Weaknesses:**

### Strength

- Evaluating self-supervised representations efficiently is an important problem. Fine-tuning them for downstream tasks can be computationally intensive, and thus, a good proxy is needed to estimate their performance.
- The proposed CLID metric demonstrates a high correlation with the classification accuracy, outperforming the baselines in terms of correlation coefficients.
- The CLID metric can be extended to evaluate out-of-distribution generalization and can be transferred to other tasks from self-supervised datasets.


### Weakness & Questions

[W1] Other pre-trained models\
The paper only demonstrates the results on supervised and joint-embedding types of self-supervised learning. Does the proposed metric work for other pre-training methods, such as reconstruction-based self-supervised learning (e.g., MAE) and vision-language pre-training (e.g., CLIP)? Adding those checkpoints to the plot of Figure 1 would be interesting and informative.

[W2] Scale of CL vs. ID\
The final CLID metric is simply a sum of two metrics. However, it is nonsensical since their scales are not calibrated. It would be great if the revised paper adds some discussion and justification that the scales of the two metrics are comparable, at least empirically, in Section 2.2. The paper partly addresses this issue through W-CLID, but the estimated weights are not discussed thoroughly.

[W3] Additional related works\
There is a line of work studying the quality of representation outside the self-supervised learning literature. These papers mostly focus on the generalizability of learned representations, often under the lens of the generalization bound in learning theory perspective. Intrinsic dimension was also studied from this perspective, to the best of my knowledge [1]. I suggest that the authors discuss such a line of work and provide a broader view for the evaluation of representations.\
On the other hand, there is learning theory literature for self-supervised learning, such as [2]. These papers provide a natural evaluation metric for self-supervised representation, and adding a discussion with them could provide a comprehensive view of the related fields.

[1] Measuring the Intrinsic Dimension of Objective Landscapes, ICLR'18.\
[2] A Theoretical Analysis of Contrastive Unsupervised Representation Learning, ICML'19.


### Comments on the resubmission

The main concerns in the last round were (1) the dependency of the CLID metric on data processing order and (2) the choice of hyperparameters. To the best of my understanding, the resubmitted paper addressed these issues by (1) making slight modifications to their method (from sequential to average accuracy) and (2) adding a sensitivity analysis of hyperparameters (Sec. 3.4). Thus, I believe the concerns from the last round have been reasonably addressed.

---

> ### Author Response · Authors · 2023-08-31
> **Thank you for your time and feedback.**
>
> W1
>
>  > Does the proposed metric work for other pre-training methods, such as  such as reconstruction-based self-supervised learning (e.g., MAE) and vision-language pre-training (e.g., CLIP)?
>
> Our work specifically focuses on JE-SSL methods.  However, your question is interesting. As a response,  we are currently  investigating the application of our metric on CLIP checkpoints.  As this may go beyond the paper's primary scope,  we would not re-compute all correlations coefficients, but we could add the results to the plot of Figure 1 as you suggest.
>
> W2. We do standardize the two metrics before summing them, as we mention this on p4: "we simply add up these values after standardization."  We propose to make this more explicit in the formula so as to remove any ambiguity.
>
> W3. Thank you for pointing out these references. We will add a paragraph in the related work section to discuss these.

---

### Review · Reviewer_ZkU8 · 2023-07-26

**Summary Of Contributions:**

This paper addresses the problem of evaluating SSL methods without requiring the proxy evaluation tasks. It argues that the combination of expressiveness and learnability can evaluate the representation of SSL methods. In particular, it uses the Intrinsic Dimension (ID) to assess expressiveness and introduce Cluster Learnability (CL) to assess learnability. The proposed indicator CLID that combines ID and CL better correlates with in-distribution model performance than other competing recent evaluation schemes on ImageNet dataset. CLID can also serve as a predictor of the transfer performance of SSL models on several visual classification tasks.

**Audience:**

Yes

**Broader Impact Concerns:**

No impact concerns.

**Claims And Evidence:**

Yes

**Requested Changes:**

1.More discuss to related work, e.g, Garrido et al. (2022) and the missing ICML 2022 paper;

2. Experimental comparison to Garrido et al. (2022).

**Strengths And Weaknesses:**


**Strengths:**

This paper is well written and I am glad to read this paper.  The topic this paper investigates is very important. I believe the researcher in SSL community would like to have a good predictor (indicator) to indicate how well the SSL methods are, without requiring the proxy evaluation tasks. The clarification of expressiveness and learnability is clear, and it is also clear that the combination of expressiveness and learnability correlates the performance of SSL methods. This paper carefully makes the claims that are well supported by the evidences.



**Weaknesses:**

1. This paper should provide more discussion to the closely related work Garrido et al. (2022). The work in Garrido et al. (2022) is simpler and more intuitive to evaluate the representation of SSL without performing the proxy tasks, as to me. Besides, this paper should also conduct experiments to compare Garrido et al. (2022) (e.g., considering the $\alpha$ in (0, 1) for Pearson coefficient). This paper should also take credits to [1], which is closely related to Garrido et al. (2022).

2. It is better to illustrate the intuition how the proposed method (by Facco et al. (2017)) in estimating the ID.


**Ref:**
[1] Exploring the Gap between Collapsed & Whitened Features in Self-Supervised Learning. ICML 2022

---

> ### Author Response · Authors · 2023-08-31
> **Thank you for your positive feedback!**
>
> >  more discussion to the closely related work Garrido et al. (2022)
>
> We appreciate your observation regarding Garrido et al. (2022) (cited in our related work section),  which is indeed closely aligned with our work. It's important to clarify that this work is contemporaneous with ours. Specifically, their paper was made available online four months *after* ours (October 5 versus June 2, 2022). We will communicate the non-anonymous details to the Action Editor.
>
> Now, we agree that the comparison of results from both papers would be insightful. While their paper employs the spectral rank of the representation as a metric, akin to measuring effective dimension, our paper approaches the problem from the perspective of performance across a tradeoff between effective dimension and learnability. Analyzing our results in light of this difference presents an intriguing avenue for further exploration.
>
> Thank you also for pointing out He & Ozay (2022).  The paper studies the spectral decay of SSL representations and its relation to generalization, which is indeed tightly related to Garrido et al. (2022),  particularly from a mathematical standpoint. We also note that this work was made available online almost one month after ours, on June 28. We will include this reference in our related work section.
>
> > illustrate the intuition how the proposed method (by Facco et al. (2017)) in estimating the ID
>
> We acknowledge that our description of this method on p4 is quite formal. We will make sure to expand the corresponding paragraph with intuitive explanations of how the method works.

---

> > ### Comment · Reviewer_ZkU8 · 2023-09-09
> > **Concerns on discussion/comparison to related work**
> >
> > I thank the response of the authors.  In my previous opinion, I mention the method of RankMe (Garrido et al. (2022).) should be compared, considering Rankme is the recent (maybe SoT) method in this topic and the submission of this paper is on June 2023. While the authors point out that this paper is originally online (arxiv) on June 2, 2022, I think this paper should compare to the paper [2] (RankMe and this paper is closely related to [2]) that is online (arxiv) on Feb 22, 2022. Besides, I think this paper miss the reference of [2].
> >
> >
> > [2] Ghosh et al, Investigating Power laws in Deep Representation Learning

---

> > > ### Author Response · Authors · 2023-09-09
> > > **Thanks for this new reference!**
> > >
> > > Gosh et al. (2022) is indeed closely related to the other references you mentioned, which focus on evaluating SSL representations through eigenspectrum decay. Specifically, assuming an eigenspectrum following a power law $\lambda_j \sim j^{-\alpha}$, the findings of Gosh et al. (2022) suggest that downstream performance improves as $\alpha$ approaches 1. It's worth noting that Garrido et al. (2022) present observations in their appendix E that appear to challenge this assertion.
> > >
> > > These studies are certainly pertinent, and we will ensure to cite and discuss them in our related work section. We'd like to emphasize how our approach differs from these methodologies. Properties like eigenspectrum decay, effective rank, or intrinsic dimension are all measures of expressiveness. In contrast, our metric combines expressiveness with learnability. Our results indicate that effective representations result from a good  tradeoff between the two.

---

### Review · Reviewer_EeFB · 2023-08-06

**Summary Of Contributions:**

The authors propose CLID, a method and metric for evaluating self supervised learning models (and potentially general neural network models), in terms of their learnability and expressivity. Those are measured using Intrinsic Dimension, a form of compressibility measure, and, Cluster Learnability, where data points encoded by a model, are clustered using K-means, and then a fitted on the training set using a KNN classifier. The resulting performance of this KNN classifier on ImageNet is how CL is obtained.

The authors conduct two empirical studies; one where they compare the CLID performance of a given model with the validation set performance of said model. This is referred to as 'in-distribution' performance. CLID is demonstrated to correlate very well with the ImageNet validation set results. The second study is 'out-of-distribution' and involves correlating the CLID performance with model performances on downstream tasks. Again, the CLID metric seems to correlate very well with the downstream task metrics, outperforming in general the other contenders chosen by the authors to appear in the table.



**Audience:**

Yes

**Claims And Evidence:**

Yes

**Requested Changes:**

The writing of the paper feels rather voluminous, and could, and ideally should be made more compact. It'll make for a paper that is easier to read and understand.

On the tables and figures where correlations are shown, the authors should explicitly state how each CLID metric was obtained (i.e. from ImageNet k-means or for per-dataset k-means etc). Table 6 is rather confusing, I am not sure what the bottom table is trying to communicate. Also, the few-shot results should ideally be in the main manuscript as they seem too important to omit, and they improve the credibility of the method introduced.

**Strengths And Weaknesses:**

Strengths:

1. The method is simple and is relatively cheap computationally.
2. It seems that the method correlates well with within and without domain generalization.
3. Authors do a fairly good job at evaluating their proposed method.
4. The method is a great way to approximate model transferability to downstream tasks, and does not require any labels or training be done to obtain.

Weaknesses:

1. It's unclear whether the CLID metric correlates as well with other downstream tasks, such as zero-shot retrieval, semantic segmentation, relational reasoning etc. While the authors provide enough empirical evaluation to demonstrate the usefulness of the method, it's not clear whether this method is truly as generally applicable as other existing methods.
2. CLID's main usefulness is that it's useful in situations where labels for a downstream task are not available as a model selector and as a measure of transferability. However, one could instead use a variety of known datasets/tasks with available labels and their transfer performance as an alternative metric -- something which is generally and extensively done today. While CLID is simply, and seemingly computationally expensive, I would still feel less confident using it in a production system as a predictor of downstream performance when compared to using a diverse set of tasks, and domains to evaluate my models and use that as a transferability measure.
3. It's also not clear whether the CLID metric is purely computed on ImageNet, or computed on each of the datasets that are being compared one by one -- this is a clarity issue that should be addressed by the authors.
4. Related to 3, the authors don't discuss whether CLID is dependent on ImageNet being the dataset to compute it on, and whether it correlates as well when the K-means cluster is computed over representations of other datasets.

---

> ### Author Response · Authors · 2023-08-31
> **Thank you for your feedback!**
>
> >  It's unclear whether the CLID metric correlates as well with other downstream tasks such as zero-shot retrieval, semantic segmentation, relational reasoning, etc.
>
> Broadening the scope to encompass a wider array of downstream transfer tasks would certainly offer valuable insights. While we acknowledge that our current setup is standard yet not exhaustive, we think it effectively highlights the relevance of our metric in evaluating out-of-domain representation performance.
>
> > However, one could instead use a variety of known datasets/tasks with available labels and their transfer performance as an alternative metric
>
> Absolutely.  In fact, almost all existing tuning and evaluation methods for SSL models rely on having labels of the considered datasets. However,  we view this as a strong limitation, especially for domains with scarse labels or high costs associated with  fine tuning for model selection. Our paper specifically addresses the performance evaluation of SSL representations in scenarios where no labels are accessible. We perceive this as significant and important problem to tackle.
>
> > It’s also not clear whether the CLID metric is purely computed on ImageNet,
>
> It is! We will make sure to make this clear in the manuscript, including in table and figures  captions as you suggest.
>
> > the authors don’t discuss whether CLID is dependent on ImageNet being the dataset to compute it on
>
> You are correct in noting that we exclusively consider ImageNet as the source dataset for computing CLID. Our focus is on ImageNet given that it covers a broad spectrum of standard SSL methods.
>
> > Table 6 is rather confusing, I am not sure what the bottom table is trying to communicate.
>
> This is in fact the continuation of the top table. It shows  correlations results for three additional baseline metrics, which are all described in Section 3.2. We acknowledge that the layout is not satisfactory, and will find a way to fusion top and bottom into a single table.  Also, we agree with your comment on few shot results and will move them to main manuscript.

---

### Decision · Action_Editors · 2023-09-25

**Recommendation:** Accept as is

**Comment:**

I want to thank the reviewers and authors for participating in the process.  I'm sorry for my part in how delayed the process played out this time.

**Audience:**

The paper is clearly of interest to those in the TMLR audiencem, proposing a way to measure performance of SSL methods.

**Claims And Evidence:**

The consensus amongst the reviewers is that the paper does provide accurate, convincing and clear evidence for its claims.